# Increasing the Amounts of Bioactive Components in American Ginseng (*Panax quinquefolium* L.) Leaves Using Far-Infrared Irradiation

**DOI:** 10.3390/foods13040607

**Published:** 2024-02-17

**Authors:** Xuan Wang, Myungjin Kim, Ruoqi Han, Jiarui Liu, Xuemei Sun, Shuyang Sun, Chengwu Jin, Dongha Cho

**Affiliations:** 1School of Food Engineering, Yantai Engineering Research Center of Green Food Processing and Quality Control, Ludong University, Yantai 264025, China; 2022121014@m.ldu.edu.cn (X.W.); 2023120979@m.ldu.edu.cn (R.H.); 2021120863@m.ldu.edu.cn (J.L.); 2875@ldu.edu.cn (X.S.); sysun81@ldu.edu.cn (S.S.); 2College of Biomedical Science, Kangwon National University, Chuncheon 24341, Republic of Korea; gmrtka@kangwon.ac.kr

**Keywords:** American ginseng leaves, far-infrared irradiation, polyphenols, ginsenosides, antioxidant activity

## Abstract

Both the roots and leaves of American ginseng contain ginsenosides and polyphenols. The impact of thermal processing on enhancing the biological activities of the root by altering its component composition has been widely reported. However, the effects of far-infrared irradiation (FIR), an efficient heat treatment method, on the bioactive components of the leaves remain to be elucidated. In the present study, we investigated the effects of FIR heat treatment between 160 and 200 °C on the deglycosylation and dehydration rates of the bioactive components in American ginseng leaves. As the temperature was increased, the amounts of common ginsenosides decreased while those of rare ginsenosides increased. After FIR heat treatment of American ginseng leaves at an optimal 190 °C, the highest total polyphenolic content and kaempferol content were detected, the antioxidant activity was significantly enhanced, and the amounts of the rare ginsenosides F4, Rg6, Rh4, Rk3, Rk1, Rg3, and Rg5 were 41, 5, 37, 64, 222, 17, and 266 times higher than those in untreated leaves, respectively. Moreover, the radical scavenging rates for 2,2-diphenyl-1-picrylhydrazyl and 2,2′-azino-*bis* (3-ethylbenzothiazoline-6-sulfonic acid) and the reducing power of the treated leaf extracts were 2.17, 1.86, and 1.77 times higher, respectively. Hence, FIR heat treatment at 190 °C is an efficient method for producing beneficial bioactive components from American ginseng leaves.

## 1. Introduction

*Panax quinquefolium* L., also known as American ginseng, is a perennial herb from the family Araliaceae [1]. The beneficial effects of American ginseng products, such as antitumor and anti-inflammatory activities, immunity-improving effects, and blood-sugar-reducing activities, are mainly attributed to bioactive ingredients such as ginsenosides, polyphenols, and polysaccharides [2,3,4,5]. The cultivation period for American ginseng is long (harvesting is generally conducted after 4–5 years [6]), with the ginsenoside content in roots dependent on the cultivation period [7]. However, the leaves can be harvested every year since the ginsenoside content therein is less dependent on the age of the plant [8]. The activity of ginsenoside is closely related to the type, quantity, and positions of the sugar groups it carries; the fewer sugar groups in ginsenoside, the higher its physiological activity [9,10]. It has been proven that the ginsenoside content in American ginseng leaves is significantly higher than that in the roots [11]. In our previous research, we showed that heat treatment transforms the constant ginsenosides in ginseng leaves into rare ginsenosides with a higher physiological activity and significantly increases the total polyphenolic content (TPC) [12]. Moreover, American ginseng leaves are a high-quality source of ginsenosides, polyphenols, and other bioactive ingredients that are not only underused but also often discarded.

The most commonly used industrial methods for American ginseng processing involve physical, chemical, or biological treatment [13], with the physical method of heat treatment being the most commonly used. The outcomes from several studies have proven that heat treatment increases the amount of highly bioactive components, such as ginsenosides, polyphenols [14], and acidic polysaccharides [15], in American ginseng. Specifically, Zhang et al. [16] reported that heat treatment decreased the amounts of constant ginsenosides and increased those of rare ginsenosides.

Far-infrared irradiation (FIR) is a heat treatment method frequently used in the food industry that can penetrate plant materials, generate heat on the surface and inside the plant, and induce the decomposition of multi-chain molecular groups [17,18]. Compared to the traditional drying method, FIR drying offers advantages such as a shorter drying time, lower costs, and a more uniform temperature distribution, ensuring the quality and safety of food. However, it is important to acknowledge that FIR drying also has its drawbacks, including the generation of high heat, which can potentially lead to burns upon exposure [18]. In our previous research, we showed that FIR can significantly increase the TPC in ginseng leaves and promote ginsenoside transformations [12]. Geng et al. [19] found in their research that after infrared drying, the content of TPC in carrot slices increased, and the antioxidant activity also increased significantly. Ren et al. [20] confirmed that the antioxidant activity of ginger was improved after infrared drying and found that there was a high correlation between antioxidant activity and TPC. Rajoriya et al. [21] found that the antioxidant activity of FIR-dried apple slices increased significantly, and there was a main correlation between them and TPC. Moreover, FIR treatment of *Lycium barbarum* [22], rice [23], black rice [24], and mango [25] can significantly increase the amounts of bioactive components such as polyphenols, flavonoids, and anthocyanins. At present, research on American ginseng mainly focuses on the roots, with less focus on the leaves. Few studies have been conducted on the effect of FIR on the bioactive components and antioxidant activities in American ginseng leaves.

Since American ginseng leaves provide a high-quality source of polyphenols, ginsenosides, and other bioactive components, exploiting this resource is of utmost importance. The purpose of the present study is to determine the effects of FIR treatment on bioactive components (polyphenols, flavonoids, and ginsenosides) and the antioxidant activity of American ginseng leaves. Our approach could pave the way for the utilization of American ginseng leaves for biomedicine production and for other health-related benefits.

## 2. Materials and Methods

### 2.1. Chemicals

Organic solvents (methanol and acetonitrile) (high-performance liquid chromatography (HPLC)-grade) were purchased from Merck KGaA (Darmstadt, Germany). Folin phenol reagent was purchased from Wako Pure Chemicals (Osaka, Japan). Kaempferol and panasenoside standards were purchased from Sigma Chemical Co. (St. Louis, MO, USA). Ginsenosides were purchased from ChromaDex (Santa Ana, CA, USA) and Ambo Institute (Daejeon, Republic of Korea).

### 2.2. Sample Collection and FIR Treatment

The pretreatment process for raw materials is shown in Figure 1. Four-year-old American ginseng leaves were collected from Wendeng, Weihai, Shandong Province. American ginseng leaf samples with the following features were selected: top of the stem broadly ovoid or obovate, about 10–15 cm long and 5–6 cm wide, dark green on the surface, light green on the back, smooth, mature, and undamaged on the surface. Intact and undamaged leaves were washed with distilled water and then wiped with gauze. Fresh American ginseng leaves were fully dried in an oven (DHG-9143BS-III; CIMO, Shanghai, China) at 50 °C for 24 h and then ground with a grinder (FW100, Taisite instrument, Tianjin, China). The powder was sieved through a 200 micron sieve to obtain a uniform particle size. The powder was divided into two groups: (1) the non-FIR-treated control (Con; 0) and (2) the FIR-treated group (treatment in an FIR dryer (HKD-10; Korea Energy Technology, Seoul, Republic of Korea) for 30 min at 160 °C (FIR-160), 170 °C (FIR-170), 180 °C (FIR-180), 190 °C (FIR-190), or 200 °C (FIR-200)). The drying chamber consisted of a stainless-steel drying chamber, sample tray, centrifugal fan, and FIR heater. Two sets of FIR heaters were used, namely, one placed at the bottom of the drying chamber and the other at the top of the drying chamber. The sample tray was arranged between and parallel to the top and bottom heaters. Hot air was circulated in the drying chamber with a fan. The inlet air temperature flowing through the hot air heater was controlled with a PID controller (the accuracy was ±1 °C).

### 2.3. Bioactive Compound Analysis

#### 2.3.1. Sample Extraction

The samples were extracted according to the method of Kim et al. [26] with some modifications. First, 2 g of sample was added to 100 mL of 80% (*v*/*v*) methanol. Sample extraction was carried out in an oscillating incubator (HNYC-202T; Ounuo, Tianjin, China) at 30 °C for 24 h. The mixture was centrifuged in a centrifugator (05pr-22 centrifuge, Hitachi, Tokyo, Japan) at 500× *g* for 10 min at room temperature. Afterward, the supernatant was collected and then filtered using Whatman No. 42 filter paper (Whatman Inc., Clifton, NJ, USA). The filtrate was concentrated using a vacuum rotary evaporator at 40 °C (Eyela Co., Tokyo, Japan). The sample was then freeze-dried using a vacuum freeze dryer (Marin Christ, Landkreis Osterode, Germany) at −60 °C and 0.071 mbar vacuum pressure [27] (the concentrated filtrate was placed in the refrigerator at −80 °C for one day before freeze-drying). The sample was stored in a refrigerator at 20 °C before being used in subsequent experiments.

#### 2.3.2. TPC Determination

The TPC was determined by using the Folin-Ciocalteu method as follows. First, 1.9 mL of distilled water and 1.0 mL of Folin-Ciocalteu reagent were mixed in a tube, after which 0.1 mL of the sample solution (2 mg/mL dissolved in 80% (*v*/*v*) methanol) was added. Subsequently, 1.0 mL of 20% Na_2_CO_3_ was added, followed by incubation for 2 h at 25 °C. Afterward, the absorbance of the sample was recorded at 765 nm with a spectrophotometer (UV-2600i, Shimadzu, Kyoto, Japan). The results are expressed as milligrams of gallic acid equivalent (mg_GAE_) per gram of dry weight.

#### 2.3.3. HPLC Analysis of Panasenoside, Kaempferol, and Ginsenoside Amounts

The amounts of panasenoside and kaempferol in American ginseng leaves were determined by using HPLC. The equipment was an HPLC system (CBM-20A; Shimadzu Co, Ltd., Kyoto, Japan) with two gradient pump systems (LC-20AT; Shimadzu, Japan), an auto-sample injector (SIL20A; Shimadzu), a UV detector (SPD-20A; Shimadzu), and a column oven (CTO-20A; Shimadzu). Ground leaves (0.1 mg) were suspended in 1 mL of 80% (*v*/*v*) methanol and filtered through a 0.22 μm membrane filter before being injected onto the HPLC system. HPLC separation was performed on an Inertsil ODS-SP C18 column (250 mm × 4.6 mm, 5 µm; GL Sciences, Tokyo, Japan). The injection volume was 10 μL. The gradient running phase was programmed as a combination of solvent A (water with 0.1% trifluoroacetic acid) and solvent B (acetonitrile), during which solvent B was sequentially increased from 14% to 18% (0 to 10 min), 18% to 30% (10 to 20 min), 30% to 60% (20 to 30 min), 60% to 65% (30 to 33 min), 65% to 100% (33 to 40 min), 100% to 100% (40 to 50 min), and then finally adjusted from 100% to 14% (50 to 65 min). The operating temperature was set to 35 °C. The flow rate of the mobile phase was kept at 1.0 mL per minute. The detector was set at 355 nm to monitor panasenoside and kaempferol.

The prepared sample solution was filtered through a 0.22 μm membrane filter. The volume injected onto a Kinetex C18 column (100 mm × 4.6 mm, 2.6 µm; Torrance, CA, USA) was 10 μL. The gradient running phase was programmed as a combination of solvent A (water) and solvent B (acetonitrile), during which solvent B was sequentially increased from 17% to 23% (0 to 30 min), 23% to 24% (30 to 35 min), 24% to 32% (35 to 45 min), 32% to 44% (45 to 48 min), 44% to 44% (48 to 52 min), 44% to 55% (52 to 65 min), 55% to 100% (65 to 85 min), 100% to 100% (85 to 95 min), and finally adjusted from 100% to 17% (95 to 105 min).

### 2.4. Determination of the Free Radical Scavenging and Antioxidant Activities

#### 2.4.1. Determination of the Free Radical Scavenging Activity

American ginseng leaf powder (90 mg) was added to 30 mL of 80% (*v*/*v*) methanol in a 50 mL tube and processed at 30 °C for 30 min via sonication (PS-40A; JeKen, Dongguan, China). Subsequently, the sample was centrifuged (J-E; Beckman Coulter Inc., Shanghai, China) at 3500 revolutions/min for 15 min. The supernatant was collected and filtered through a 0.22 μm membrane filter.

The DPPH free radical scavenging activity was measured using the procedure described by Kossah et al. [28] with some modifications. Briefly, 1 mL of sample solution at varying concentrations was mixed with 3 mL of DPPH solution. After reacting in the dark at room temperature for 30 min, the absorbance was measured at 517 nm.

The radical scavenging activity of ABTS was measured using the procedure described by Lim et al. [29] with some modifications. Briefly, 0.5 mL of the sample solution at varying concentrations was mixed with 5 mL of ABTS solution. After reacting in the dark at room temperature for 10 min, the absorbance was measured at 734 nm.

#### 2.4.2. Determination of the Reducing Power

Sample processing followed the same procedure as that mentioned in Section 2.4.1. Measurement of the reducing power was conducted using the procedure described by Oyaizu et al. [30] with some modifications. Briefly, 0.4 mL of sample solution at varying concentrations was mixed with 1 mL of 0.2 mol/L phosphate buffer (pH 6.6) and 1 mL of 1% potassium ferrocyanide solution. The mixture was then incubated in a 50 °C water bath for 30 min, after which 1 mL of 10% trichloroacetic acid was added, followed by thorough mixing and centrifugation at 3000 revolutions/min for 10 min. After centrifugation, 2 mL of the supernatant was removed and 2 mL of distilled water and 0.4 mL of 0.1% ferric chloride were added. Next, the mixture was vortexed, and the absorbance of the mixture was measured at 700 nm.

### 2.5. Statistical Analysis

The data were analyzed statistically using SAS software (Enterprise Guide version 7.1; SAS Institute Inc., Cary, NC, USA). The data from the non-FIR-treated control and FIR treatment groups were compared by using a one-way analysis of variance. Differences between the experimental groups were evaluated by using Tukey’s honestly significant difference (HSD) test at the *p* < 0.05 significance level.

## 3. Results and Discussion

### 3.1. The Effect of FIR Heat Treatment on the TPC in American Ginseng Leaves

The change in TPC in American ginseng leaves after FIR heat treatment is shown in Figure 2. The TPC gradually increased with the treatment temperature up to 190 °C and then declined. At 190 °C, the highest TPC was 15.89 mg/g, which was 1.56 times higher than that without treatment. According to Hwang et al. [31], phenolic compounds can scavenge free radicals by providing hydrogen; thus, promoting polyphenol generation in American ginseng leaves via FIR heat treatment improves the antioxidant activity of the extract. Our results are similar to those reported before. For example, Jeong et al. [32] found that the content of TPC increased significantly after baking dry ginseng powder, probably because Maillard reaction products with a phenol structure were produced during baking, which increased the content of TPC. Jin et al. [33] found that the content of TPC increased significantly after baking ginseng roots at 140–200 °C and speculated that baking would transform the combined polyphenol compounds into free polyphenols to increase the content of TPC. In the present study, the TPC in American ginseng leaves increased after FIR heat treatment. We speculate that there are two reasons for this. First, most phenolic compounds in plants are covalently bonded to insoluble polymers and are released by heat treatment [34,35]. Second, heating can promote the hydrolysis of macromolecular polyphenols, thereby producing smaller molecular polyphenols [36]. From the above experimental results, it can be concluded that the highest TPC content in American ginseng leaves was produced via FIR heat treatment at 190 °C.

### 3.2. The Effect of FIR Heat Treatment on the Amounts of Panasenoside and Kaempferol in American Ginseng Leaves

Figure 3 shows the kaempferol and panasenoside content in American ginseng leaves before and after FIR treatment at various temperatures. The panasenoside content in the leaves before FIR treatment was 18.52 mg/g. Increasing the FIR treatment temperature decreased the panasenoside content, with the lowest value observed after treatment at 200 °C (4.49 mg/g), which was 76% lower than that without treatment. Contrary to the change in panasenoside content, the kaempferol content increased from 0.026 mg/g without treatment up to 0.841 mg/g (FIR-190) at 190 °C, after which it decreased. The kaempferol content after FIR treatment at 190 °C was 32 times higher than that without treatment.

Panasenoside (kaempferol 3-o-glucosyl-(1→2)-galactoside) is a kaempferol glycoside [37]. According to the findings of our previous research on ginseng leaves, we speculate that during FIR treatment, energy can cause panasenoside deglycosylate to break the bond between kaempferol aglycone and the first glycosyl, resulting in a significant increase in kaempferol content. However, astragalin (kaempferol 3-β-D-glucopyranoside) was not detected. The deglycosylation of panasenoside is concurrent with kaempferol degradation; increasing the FIR temperature gradually caused the panasenoside deglycosylation rate to become lower than the kaempferol degradation rate. This explains why the panasenoside content in American ginseng leaves decreased rapidly while the kaempferol content increased relatively slowly, reaching the highest value after FIR treatment at 190 °C [12]. Although the kaempferol content in American ginseng leaves reached the highest value after FIR treatment at 190 °C, its content in ginseng leaves was different, and it stopped increasing after FIR treatment at 180 °C. This could have been caused by the different compositions and amounts of panasenoside in the raw material used. Kaempferol (a deglycosylation product) has high anti-inflammatory [2], antitumor [3], and antioxidant activities [38]. According to the above experimental results, it can be concluded that the kaempferol content in American ginseng leaves was markedly increased by FIR treatment at 190 °C.

### 3.3. The Effect of FIR Treatment on the Ginsenoside Content in American Ginseng Leaves

#### 3.3.1. Protopanaxadiol (PPD)-Type Ginsenosides

The change in PPD-type ginsenoside content in American ginseng leaves after FIR heat treatment at various temperatures is shown in Figure 4A and Table 1. The amounts of the main ginsenosides (Rb1, Rb2, Rb3, Rc) were negatively correlated with the FIR treatment temperature. When the FIR temperature was 200 °C, their amounts (2.12, 4.10, 12.70, 1.70, and 8.66 mg/g, respectively) were even lower than in the untreated leaves (equating to reductions of 73%, 68%, 74%, 64%, and 73%, respectively). On the contrary, the amounts of rare PPD-type ginsenosides (Rg5, Rk1, and Rg3) showed an upward trend with increasing FIR temperature; their amounts were the highest after FIR treatment at 190 °C (13.37, 4.46, and 1.94 mg/g compared to 0.05, 0.02, and 0.11 mg/g in untreated leaves, equating to 266-, 222-, and 17-fold increases, respectively).

The conditions applied can be dry heat, wet heat, etc. The main pathways for PPD-type ginsenoside conversion during dry heat treatment are as follows: Rb1, Rb2, Rb3, Rc, Rd→Rg3→Rk1, and Rg5 [39]. Increasing the FIR temperature significantly increased the amounts of Rk1, Rg5, and the other ginsenosides but not Rg3. This could be because Rg3, which is an intermediate produced by the deglycosylation of Rb1, Rb2, Rb3, Rc, Rd, and other ginsenosides, can also undergo dehydration and deglycosylation at similar reaction rates. Although ginsenosides have great potential, the bioavailability of the main ginsenosides is low because their absorption rates in the circulatory system are very low. On the contrary, rare ginsenosides exhibit good bioavailability because they can more easily permeate through cell membranes due to their small molecular weight [40]. In the present study, increasing the FIR treatment temperature decreased the main PPD-type ginsenoside content by 72% and increased the rare ginsenoside content hundreds of times compared to its value without treatment. Although many researchers have reported that heat treatment can promote ginsenoside transformations, different heat treatment methods and conditions affect their transformation rate [40,41,42]. For example, Park et al. [43] reported that after heat treatment at 120 °C, the main ginsenosides were converted into rare ginsenosides (Rg3, Rg5, and Rk1) with a lower polarity; they were difficult to detect in untreated samples and increased to 15.2, 3.6, and 2.9 µg/mg, respectively, after treatment. Pu et al. [44] reported that the rare ginsenosides Rg3, Rk1, and Rg5 could be produced effectively after the ginseng was expanded. Kim et al. [45] reported that the content of ginsenoside changed significantly with steaming temperature. With the increase in temperature, the contents of the ginsenosides Rg3, Rk1, and Rg5 increased from 0.69, 1.11, and 1.79 mg/g to 4.08, 5.16, and 8.98 mg/g, respectively. We found that the amounts of the rare ginsenosides Rg5 and Rk1 were higher, while the Rg3 content was lower than they reported. This could be because the reaction rate of Rd→Rg3 deglycosylation was different to that of Rg3→Rg5 and Rk1 dehydration due to the differences in the heat treatment method and temperature. Steam and drying treatments can increase the Rg3, Rg5, and Rk1 amounts and decrease the Rb1 amount [46]. The findings from the above research are similar to ours for FIR treatment of American ginseng leaves. When increasing the FIR treatment temperature from 160 to 190 °C, the amounts of PPD-type rare ginsenosides increased significantly. In particular, the amount of the rare ginsenoside Rg5 increased by 1.76 mg/g (con→FIR-160), 2.10 mg/g (FIR-160→FIR-170), 4.31 mg/g (FIR-170→FIR-180), and 5.16 mg/g (FIR-180→FIR-190). Heat treatment can transform the main PPD-type ginsenosides into rare ones with higher physiological activities and a higher clinical value.

#### 3.3.2. Protopanaxatriol (PPT)-Type Ginsenosides

Changes in the amounts of PPT-type ginsenosides in American ginseng leaves according to the FIR treatment temperature shown in Figure 4B and Table 2 are similar to those observed for PPD-type ginsenosides; i.e., the main ginsenoside content decreased and the corresponding rare ginsenoside content increased with increasing temperature. When increasing the FIR treatment temperature, the amounts of the main ginsenosides Re and Rg1 decreased while those of Rg2 and Rh1 increased first and then decreased. Moreover, after FIR treatment up to 190 °C, the amounts of the rare ginsenosides F4, Rg6, Rh4, and Rk3 did not increase (1.89, 1.82, 1.07, and 0.61 mg/g, respectively). Compared with untreated leaves, the amounts of the main ginsenosides Re and Rg1 decreased by 79.27% and 77.96%, respectively, while those of the rare ginsenosides Rk3, F4, Rh4, and Rg6 were 64, 41, 37, and 5 times higher, respectively.

Under FIR heat treatment, the main transformation paths for PPT-type ginsenosides are as follows: Re→Rg2→Rg6 and F4 and Rg1→Rh1→Rh4 and Rk3 [39]. Our experimental results show that FIR treatment increases the amounts of the rare ginsenosides Rg6, F4, Rh4, and Rk3 by 4-, 40-, 36-, and 63-fold, respectively. Xue et al. [8] used an autoclave to heat American ginseng to a high temperature and obtained similar results. With increasing temperature and time, the PPT-type rare ginsenoside content in American ginseng also showed an increasing trend. Ji et al. [47] reported that processing a ginseng extract at 90 °C greatly increased the amounts of the rare ginsenosides F4, Rk3, and Rh4 compared to the untreated ginseng extract. Their results are similar to ours in that PPT-type main ginsenosides were transformed into rare ginsenosides via heat treatment. Rg2 and Rh1 are intermediate products of the transformation of Re and Rg1, respectively. Increasing the FIR treatment temperature caused the amounts of the Rg2 and Rh1 ginsenosides to first increase and then decrease, which could be because the deglycosylation rate of transformation of Re and Rg1 into Rg2 and Rh1, respectively, is lower than the dehydration rate of the latter two products [48]. As shown in Figure 4A,B, the highest amounts of rare PPD- and PPT-type ginsenosides were obtained at an FIR treatment temperature of 190 °C and decreased when the temperature was increased to 200 °C; this could have been caused by ginsenoside degradation due to the higher FIR energy [12]. However, the decreases in rare ginsenoside contents at 200 °C are slightly different from that after FIR treatment of ginseng leaves [12], which may have been caused by the different compositions and amounts of ginsenosides in the raw materials [49]. Although increasing the FIR treatment temperature from 160 to 190 °C increased rare PPT-type and PPD-type ginsenosides to different extents, the change in content increases was not significant. This could be due to the different structures of PPD-type and PPT-type ginsenoside aglycones, resulting in different dehydration rates.

### 3.4. The Effects of FIR Treatment on the Free Radical Scavenging and Antioxidant Activities of American Ginseng Leaves

2,2-Diphenyl-1-picrylhydrazyl (DPPH) and 2,2′-azino-bis (3-ethylbenzothiazoline-6-sulfonic acid) (ABTS) scavenging assays and the ferric reducing antioxidant power (FRAP) assay were used to determine the effect of FIR treatment on the antioxidant capacity of American ginseng leaf extracts. The results shown in Figure 5 indicate that FIR treatment significantly improved the antioxidant capacity of American ginseng leaf extracts. The DPPH and ABTS scavenging activities of American ginseng leaves treated with FIR at 190 °C were 2.17 times (32.08% vs. 69.62%) and 1.86 times (35.88% vs. 66.61%) higher than those in untreated leaves. The absorbance measured using the FRAP method increased 1.77-fold (0.212 vs. 0.374) after FIR treatment at 190 °C. When the FIR treatment temperature was raised to 200 °C, the DPPH and ABTS scavenging rates and the reducing power all exhibited a downward trend.

Many antioxidants naturally covalently bind to insoluble polymers. If the bonds are weak, FIR can release and activate natural low-molecular-weight antioxidants such as phenolic acids, flavonoids, and carotenes [50]. The results in Figure 2 and Figure 5 indicate that FIR treatment up to 190 °C increases the TPC and antioxidant activity in American ginseng leaves, after which they both decrease. These findings are similar to our previous research results in ginseng leaves. We speculate that FIR treatment can improve the antioxidant activity of ginseng leaves by releasing bound phenols. When the FIR treatment temperature is too high, phenolic degradation is accelerated, resulting in decreases in the TPC and antioxidant activity [12]. Similar results have also been reported after FIR treatment of rice, peanut shells, and *Angelica sinensis* [23,50,51]. Although many studies have been conducted on American ginseng leaves, this is the first time that the antioxidant activity of American ginseng leaf extracts has been assessed after FIR treatment. The rational use of FIR treatment could improve the antioxidant activity of medicinal plants and increase their medicinal value. From the above experimental results, it can be concluded that 190 °C is the optimal FIR treatment temperature for processing American ginseng leaves to obtain a higher antioxidant activity.

## 4. Conclusions

In this study, the effects of FIR on bioactive components and the antioxidant activity of American ginseng leaves were investigated in detail. Our research results show that FIR treatment is an efficient processing method for producing beneficial bioactive components in American ginseng leaves. After FIR treatment at 190 °C, the highest TPC was 1.56 times higher compared to the untreated samples. The kaempferol content after FIR treatment at 190 °C was 32 times higher than that of the untreated samples. The contents of the ginsenosides Rk3, F4, Rh4, Rg6, Rg5, Rk1, and Rg3 were 64, 41, 37, 5, 266, 222, and 17 times higher, respectively, compared to the untreated group. Moreover, the antioxidant activity was significantly improved with an increase in the TPC. The results of this study could provide a reference for the rationalization and utilization of high-value biomedicines from American ginseng leaves. (In the following research, we will explore whether there is a direct correlation between the changes in ginsenoside content and antioxidant activity and try to analyze its mechanism.)

## Figures and Tables

**Figure 1 foods-13-00607-f001:**
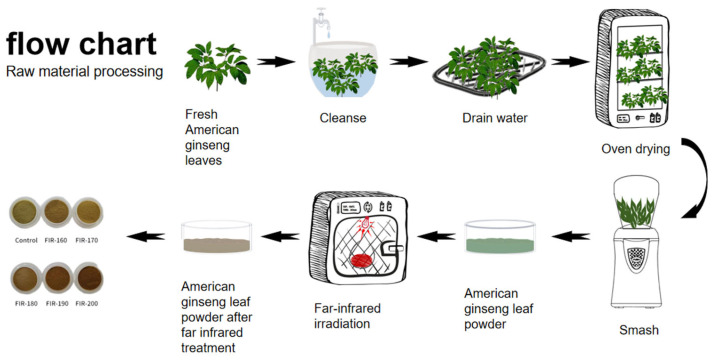
Flow chart of raw material pretreatment.

**Figure 2 foods-13-00607-f002:**
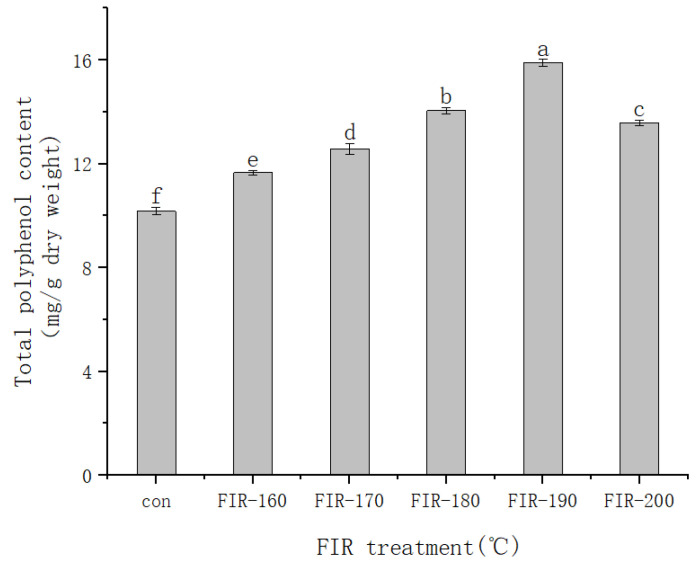
Changes in the total polyphenol content (TPC) of American ginseng leaves treated with FIR at various temperatures. The values are expressed as the mean and the standard error (*n* = 3). Letters a–f above the bar graphs indicate significant differences at *p* < 0.05 using Tukey’s honestly significant difference (HSD) test.

**Figure 3 foods-13-00607-f003:**
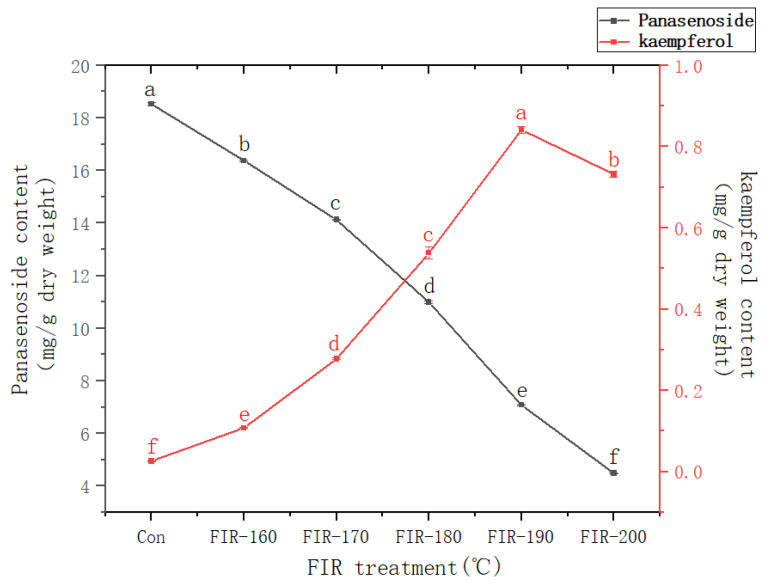
Changes in panasenoside and kaempferol contents in American ginseng leaves treated via FIR at various temperatures. The values are expressed as the mean and the standard error (*n* = 3). Letters a–f above the line charts indicate significant differences at *p* < 0.05 using Tukey’s HSD test.

**Figure 4 foods-13-00607-f004:**
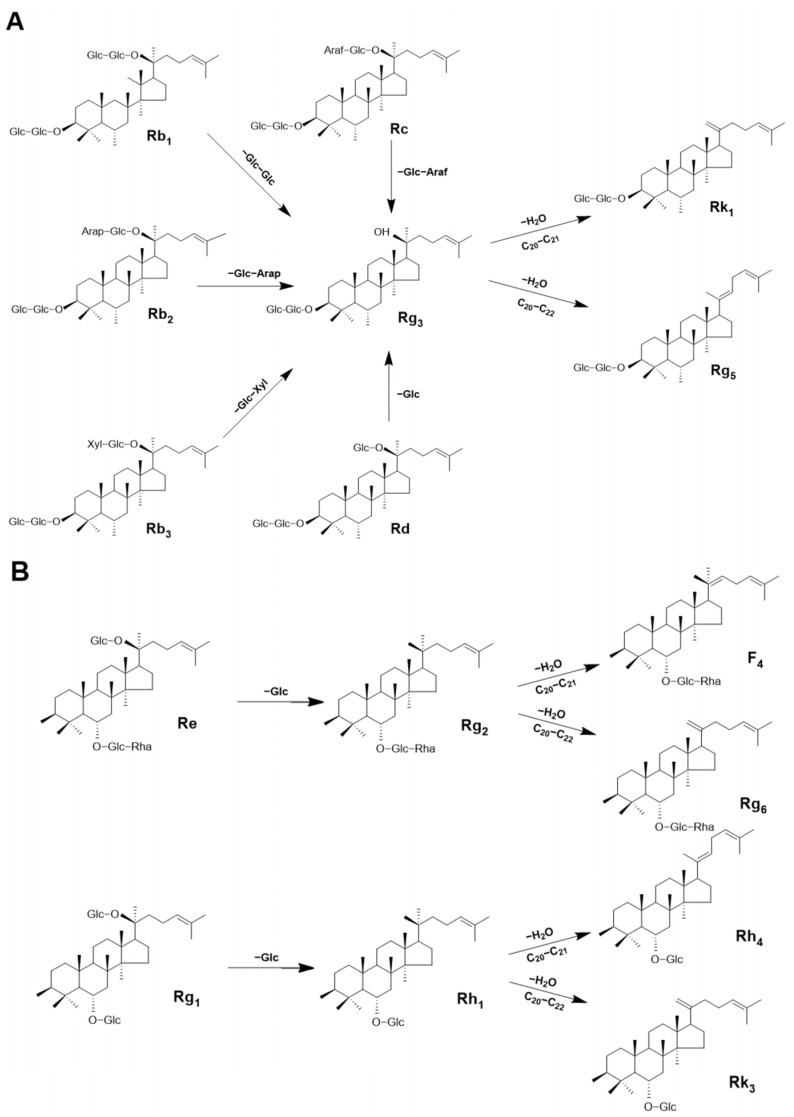
Schematic diagram of the transformation of (**A**) protopanaxadiol (PPD) and (**B**) protopanaxatriol (PPT) ginsenosides in American ginseng leaves after FIR treatment.

**Figure 5 foods-13-00607-f005:**
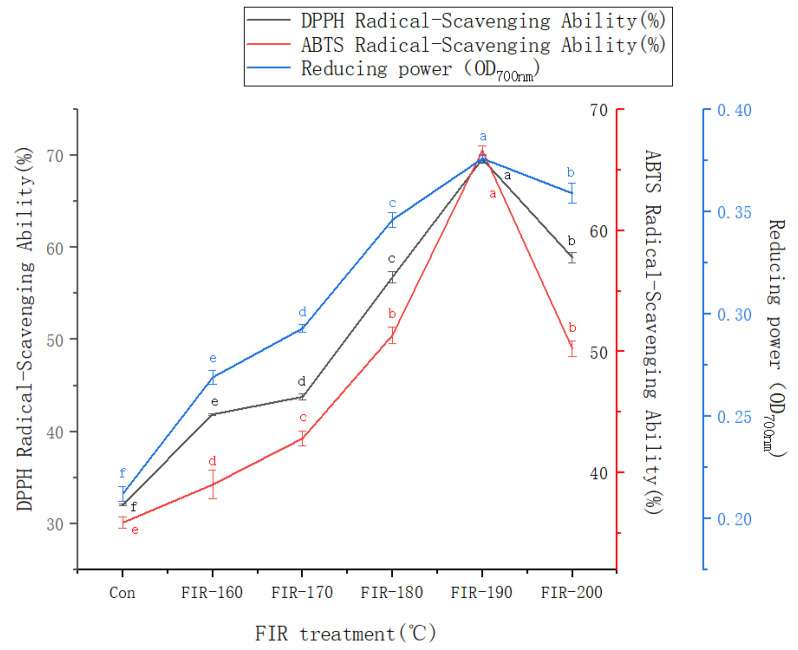
Changes in the 2,2-diphenyl-1-picrylhydrazyl (DPPH) and 2,2′-azino-bis (3-ethylbenz- othiazoline-6-sulfonic acid) (ABTS) scavenging rates and reducing power in American ginseng leaves treated with FIR at various temperatures. The values are expressed as the mean and the standard error (*n* = 3). Letters a–f above the bar graphs indicate significant differences at *p* < 0.05 using Tukey’s HSD test.

**Table 1 foods-13-00607-t001:** Changes in the content of protopanaxadiol (PPD) ginsenoside in leaves of American ginseng treated with FIR at different temperatures.

Ginsenoside (mg/g)	FIR Treatment (°C)					
	Con	FIR-160	FIR-170	FIR-180	FIR-190	FIR-200
Rb1	7.88 ± 0.03 a	6.54 ± 0.05 b	5.91 ± 0.08 c	4.86 ± 0.04 d	2.82 ± 0.02 e	2.12 ± 0.02 f
Rb2	12.75 ± 0.03 a	11.69 ± 0.03 b	11.11 ± 0.05 c	10.13 ± 0.03 d	5.43 ± 0.02 e	4.10 ± 0.02 f
Rb3	48.25 ± 0.05 a	43.53 ± 0.06 b	40.65 ± 0.05 c	30.33 ± 0.04 d	17.46 ± 0.02 e	12.70 ± 0.02 f
Rc	4.71 ± 0.03 a	4.34 ± 0.04 b	4.11 ± 0.04 c	3.60 ± 0.05 d	2.18 ± 0.02 e	1.70 ± 0.02 f
Rd	31.73 ± 0.03 a	27.48 ± 0.04 b	25.48 ± 0.05 c	20.23 ± 0.04 d	11.75 ± 0.02 e	8.66 ± 0.02 f
Rk1	0.02 ± 0.00 e	0.56 ± 0.01 d	1.20 ± 0.01 c	2.62 ± 0.02 b	4.46 ± 0.01 a	4.46 ± 0.00 a
Rg3	0.11 ± 0.00 e	0.65 ± 0.03 d	0.93 ± 0.01 c	1.45 ± 0.02 b	1.94 ± 0.02 a	1.97 ± 0.01 a
Rg5	0.05 ± 0.01 e	1.81 ± 0.03 d	3.90 ± 0.03 c	8.21 ± 0.03 b	13.37 ± 0.02 a	13.38 ± 0.01 a

The values are expressed as the mean and the standard error (*n* = 3). Letters a–f above the line charts indicate significant differences at *p* < 0.05 using Tukey’s HSD test.

**Table 2 foods-13-00607-t002:** Changes in the content of protopanaxatriol (PPT) ginsenoside in leaves of American ginseng treated with FIR at different temperatures.

Ginsenoside (mg/g)	FIR Treatment (°C)					
	Con	FIR-160	FIR-170	FIR-180	FIR-190	FIR-200
Rg1	8.12 ± 0.03 a	5.47 ± 0.07 b	5.20 ± 0.02 c	3.99 ± 0.03 d	2.40 ± 0.02 e	1.79 ± 0.02 f
Re	29.18 ± 0.03 a	21.47 ± 0.04 b	19.93 ± 0.04 c	15.15 ± 0.04 d	8.35 ± 0.02 e	6.05 ± 0.02 f
Rg2	0.77 ± 0.01 e	1.74 ± 0.01 b	1.75 ± 0.03 b	1.83 ± 0.02 a	1.65 ± 0.01 c	1.35 ± 0.02 d
Rh1	1.36 ± 0.01 c	1.50 ± 0.02 b	1.76 ± 0.03 a	0.75 ± 0.01 d	0.61 ± 0.00 e	0.57 ± 0.01 f
F4	0.05 ± 0.00 e	0.56 ± 0.01 d	0.85 ± 0.01 c	1.41 ± 0.02 b	1.89 ± 0.00 a	1.88 ± 0.00 a
Rh4	0.03 ± 0.00 e	0.35 ± 0.00 d	0.54 ± 0.00 c	0.75 ± 0.00 b	1.07 ± 0.00 a	1.07 ± 0.00 a
Rg6	0.36 ± 0.00 f	0.83 ± 0.01 e	0.99 ± 0.01 d	1.42 ± 0.01 c	1.82 ± 0.01 a	1.79 ± 0.01 b
Rk3	0.01 ± 0.00 e	0.12 ± 0.00 d	0.20 ± 0.00 c	0.38 ± 0.00 b	0.61 ± 0.01 a	0.60 ± 0.00 a

The values are expressed as the mean and the standard error (*n* = 3). Letters a–f above the line charts indicate significant differences at *p* < 0.05 using Tukey’s HSD test.

## Data Availability

The data presented in this study are available in the article.

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
