# Peer review of "Increasing the Amounts of Bioactive Components in American Ginseng (Panax quinquefolium L.) Leaves Using Far-Infrared Irradiation"

_foods, 2024, doi:10.3390/foods13040607_

Round 1

Reviewer 1 Report

Comments and Suggestions for Authors

The present work is focused on bioactive compounds enhancement in American Ginseng leaves through far infra red radiation. This work is interesting but it cannot acceptable in its current form. I suggest authors to improve the Ms.

Comments 

1. Author can add novelty of this work in Abstract.

2. Research gap is missing in Introduction section. Adding this will improve this particular section.

3. Author should mention the relevant reference for sample extraction method (section 2.3.1).

4. Authors should merge the section 2.3.3 and 2.3.4. For HPLC analysis.

5. Recent relevant studies should be added in discussion part of the Ms.

6. If possible author can present Figure 4 A, B  and C in single Figure by adding Y2 and Y3 axis.

7. Grammatical errors should be thoroughly checked.

8. References must be cross checked.

Comments on the Quality of English Language

Dear Editor, The Ms. can be considered for publication moderate English revision.

Thanks & Regards,

Author Response

Response to review comments

(Foods-2818452)

Dear editor and a reviewer,

   Thank you for giving us the opportunity to submit a revised draft of our manuscript titled Increasing the Amounts of Bioactive Components in American Ginseng Leaves by Using Far-Infrared Irradiation. We appreciate the time and effort that you and the reviewers have dedicated to provide your valuable feedback on our manuscript. We are grateful to the reviewers for their insightful comments on our paper. We have been able to incorporate changes to reflect most of the suggestions provided by the reviewers. We have highlighted the changes within the manuscript.

    Here is a point-by-point response to the reviewers’ comments and concerns.

 Reviewer 1.

ABSTRACT:

  1. - Author can add novelty of this work in Abstract.

Response: Thank you very much for your suggestions. We added novelty of this work in Abstract on lines -.

- Lines - : Both the roots and leaves of American ginseng contain ginsenosides and polyphenols. The impact of thermal processing on enhancing the biological activities of the root by altering its component composition has been widely reported. However, the effects of far-infrared irradiation (FIR), an efficient heat treatment method, on the bioactive components of the leaves remain to be elucidated.

INTRODUCTION:

  1. - Research gap is missing in Introduction section. Adding this will improve this particular section.

Response: Thank you very much for your suggestions. We added the blank part of the research on lines -.

- Lines - : At present, the research on American ginseng mainly focuses on the roots, but less on the leaves. Few studies have been conducted on the effect of FIR on the bioactive components in American ginseng leaves.

MATERIALS AND METHODS:

  1. - Author should mention the relevant reference for sample extraction method (section 2.3.1).

Response: Thank you very much for your suggestions. We added references for sample extraction.

  1. - Authors should merge the section 2.3.3 and 2.3.4. For HPLC

Response: Thank you very much for your suggestions. We merged section 2.3.3 with section 2.3.4.

RESULTS AND DISCUSSION:

  1. - Recent relevant studies should be added in discussion part of the Ms.

Response: Thank you very much for your suggestions. We added recent relevant studies in discussion part of the Ms.

  1. - If possible author can present Figure 4 A, B and C in single Figure by adding Y2 and Y3 axis.

Response: We combined three figures into one figure to show antioxidant activity.

  1. - Grammatical errors should be thoroughly checked.

Response: We chose the grammar correction institution recommended by Foods magazine to correct the grammar.

REFERENCES:

  1. - References must be cross checked.

Response: Thank you very much for your suggestions. We checked and corrected the references.

Reviewer 2 Report

Comments and Suggestions for Authors

The work is framed in the use of infrared radiation to process American ginseng leaves with a focus on bioactive compounds.

The work is interesting and well organised, however there are opportunities for improvement which are detailed below:

The title could include the scientific name of American ginseng.

Prefer the use of corporate emails rather than personal emails.

In l56 when talking about the use of infrared, some advantages and disadvantages of this type of heating in relation to traditional processing methods could be added.

In l74 it is not specified which solvent was used.

In many parts of materials and methods all the equipment used (grinder, spectrophotometer, centrifuges, sonicator, etc) are not properly specified, please check.

In the sample collection are there any characteristics of the sheets such as colour, size and shape that are relevant to standardise the experiment?

Does the infrared equipment have any special features or is it only commercially available equipment, and do you need to specify things like type of IR element, power, whether there was any circulating air flow, etc.?

It is also important to provide information on the moisture content of the leaves in their fresh, ground and then dried state for the different powers used.

Generally, when infrared radiation is used, the powers used are specified, which result in temperatures, and it is also important to mention how and where the temperatures were measured.

In the freeze-drying operation, temperatures and vacuum pressures used for this process are not mentioned.

It would be interesting if the mechanisms associated with the bioactive substances studied were mentioned in the introduction.

The graphs in figure 3 do not have units on the Y scale, it would be useful to explore another clearer way to present these results, perhaps combinations of graphs and tables.

The conclusions can be improved and expanded.

Author Response

Response to review comments

(Foods-2818452)

Dear editor and a reviewer,

   Thank you for giving us the opportunity to submit a revised draft of our manuscript titled Increasing the Amounts of Bioactive Components in American Ginseng Leaves by Using Far-Infrared Irradiation. We appreciate the time and effort that you and the reviewers have dedicated to provide your valuable feedback on our manuscript. We are grateful to the reviewers for their insightful comments on our paper. We have been able to incorporate changes to reflect most of the suggestions provided by the reviewers. We have highlighted the changes within the manuscript.

    Here is a point-by-point response to the reviewers’ comments and concerns.

 Reviewer 2.

TITLE:

  1. - The title could include the scientific name of American ginseng.

Response: Thank you very much for your suggestions. We modified the title to “Increasing the Amounts of Bioactive Components in American Ginseng (Panax quinquefolium L.) Leaves by Using Far-Infrared Irradiation”.

EMAILS:

  1. - Prefer the use of corporate emails rather than personal emails.

Response: We replaced personal email with corporate email.

INTRODUCTION:

  1. - In l56 when talking about the use of infrared, some advantages and disadvantages of this type of heating in relation to traditional processing methods could be added.

Response: Thank you very much for your suggestions. We added the advantages and disadvantages of FIR drying on lines - .

- Lines - : Compared to the traditional drying method, FIR drying offers advantages such as shorter drying time, lower costs, and more uniform temperature distribution, ensuring the quality and safety of food. However, it is important to acknowledge that FIR drying also has its drawbacks, including the generation of high heat, which can potentially lead to burns upon exposure.

  1. - It would be interesting if the mechanisms associated with the bioactive substances studied were mentioned in the introduction.

Response: We added the activity change caused by the content change of related active substances on lines - .

- Lines - : Geng et al. found in their research that after infrared drying, the content of TPC in carrot slices increased, and the antioxidant activity also increased significantly. Ren et al confirmed that the antioxidant activity of ginger was improved after infrared drying, and found that there was a high correlation between antioxidant activity and TPC. Rajoriya et al. found that the antioxidant activity of far-infrared dried apple slices increased significantly, and there was a main correlation between them and TPC.

MATERIALS AND METHODS:

  1. - In l74 it is not specified which solvent was used.

Response: We added the kinds of organic solvents used.

  1. - In many parts of materials and methods all the equipment used (grinder, spectrophotometer, centrifuges, sonicator, etc) are not properly specified, please check.

Response: Thank you very much for your suggestions. We have perfected the information of the equipment used.

  1. - In the sample collection are there any characteristics of the sheets such as colour, size and shape that are relevant to standardise the experiment?

Response: We added the basis for selecting raw materials on lines -.

- Lines - : Select the leaf samples of American ginseng with the top of the stem, broadly ovoid or obovate, about 10-15cm long and 5-6cm wide, dark green on the surface, light green on the back, smooth, mature and undamaged on the surface.

  1. - Does the infrared equipment have any special features or is it only commercially available equipment, and do you need to specify things like type of IR element, power, whether there was any circulating air flow, etc.?

Response: We added information about infrared devices on lines -.

- Lines - : The drying chamber consists of stainless steel drying chamber, sample tray, centrifugal fan and FIR heater. Two sets of FIR heaters are placed, one at the bottom of the drying chamber and the other at the top of the drying chamber. The sample tray is arranged between and parallel to the top and bottom heaters. Hot air is circulated in the drying chamber by a fan. The inlet air temperature flowing through the hot air heater is controlled by PID controller (the accuracy is 1℃).

  1. - It is also important to provide information on the moisture content of the leaves in their fresh, ground and then dried state for the different powers used.

Response: Our raw material processing flow is as follows: firstly, fresh American ginseng leaves are collected, cleaned, drained, dried in an oven, and the dried American ginseng leaves are crushed, and then the powder of American ginseng leaves is treated by far infrared irradiation. Because the leaves of American ginseng were completely dried in the oven before far infrared treatment, we thought that there was no significant difference in water content in the leaves of American ginseng, which had no obvious influence on the subsequent results, so the water content was not determined.

  1. - Generally, when infrared radiation is used, the powers used are specified, which result in temperatures, and it is also important to mention how and where the temperatures were measured.

Response: The far infrared drying equipment we use controls the temperature through radiation intensity, and the required temperature is set in the equipment program, and the equipment will automatically adjust the temperature without manual temperature measurement.

  1. - In the freeze-drying operation, temperatures and vacuum pressures used for this process are not mentioned.

Response: We added the temperature and vacuum pressure needed for freeze-drying on lines - .

- Lines - : (The sample was then freeze-dried using a vacuum freeze dryer (Christ Alpaha 1-4, Germany) at -60℃ and 0.071 mbar vacuum pressure ( The concentrated filtrate was placed in the refrigerator at -80℃ for one day before freeze-drying).

RESULTS AND DISCUSSION:

  1. - The graphs in figure 3 do not have units on the Y scale, it would be useful to explore another clearer way to present these results, perhaps combinations of graphs and tables.

Response: Thank you very much for your suggestions. We converted the graphs of ginsenoside into the form of chart combination.

CONCLUSIONS:

  1. - The conclusions can be improved and expanded.

Response: Thank you very much for your suggestions. We improved the conclusion and extended it on lines - .

- Lines - : Under the FIR treatment at 190℃, the highest TPC was 1.56 times higher compared to the untreated samples. The kaempferol content after FIR treatment at 190℃ was 32 times higher than that of the untreated samples. The contents of ginsenoside Rk3, F4, Rh4, Rg6, Rg5, Rk1, and Rg3 were 64, 41, 37, 5, 266, 222, and 17 times higher, respectively, compared to the untreated group.

In the following research, we will explore whether there is a direct correlation between the changes of ginsenoside content and antioxidant activity, and try to analyze its mechanism.

Round 2

Reviewer 2 Report

Comments and Suggestions for Authors

The authors have responded satisfactorily to all comments and suggestions, thus improving the work considerably. It is recommended that the work be accepted in its present form.